# Vision-EKIPL: External Knowledge-Infused Policy Learning for Visual Reasoning

## Abstract

Visual reasoning is crucial for understanding complex multimodal data and advancing Artificial General Intelligence. Existing methods enhance the reasoning capability of Multimodal Large Language Models (MLLMs) through Reinforcement Learning (RL) fine-tuning (e.g., GRPO). However, current RL approaches sample action [1] groups solely from the policy model itself, which limits the upper boundary of the model's reasoning capability and leads to inefficient training. To address these limitations, this paper proposes a novel RL framework called **Vision-EKIPL**. The core of this framework lies in introducing high-quality actions generated by external auxiliary models during the RL training process to guide the optimization of the policy model. The policy learning with knowledge infusion from external models significantly expands the model's exploration space, effectively improves the reasoning boundary, and substantially accelerates training convergence speed and efficiency. Experimental results demonstrate that our proposed Vision-EKIPL achieved up to a 5% performance improvement on the Reason-RFT-CoT Benchmark compared to the state-of-the-art (SOTA). It reveals that Vision-EKIPL can overcome the limitations of traditional RL methods, significantly enhance the visual reasoning performance of MLLMs, and provide a new effective paradigm for research in this field.

## 1 Introduction

Visual reasoning, a core cognitive ability involving interpretation, inference, and logical thinking based on visual information, has emerged as a critical and highly challenging research frontier within the field of Artificial Intelligence Lindström & Abraham (2022); OpenAI (2024). This capability serves as a fundamental cornerstone for numerous complex AI applications, ranging from image recognition Petrou & Petrou (2010); Ji et al. (2024) and scene understanding Cordts et al. (2016); Yang et al. (2024b) to autonomous robotic navigation Ji et al. (2025); Li et al. (2024c) and autonomous driving Hao et al. (2025b; 2024b;a;c), underscoring its growing strategic importance.

To effectively enhance the visual reasoning capability of machines, the research community has explored diverse technical approaches. Current mainstream research paradigms can be broadly categorized into three types: (1) neural-symbolic methods Garcez et al. (2019); Amizadeh et al. (2020b); Choi et al. (2024); Zhang et al. (2024a); Gupta & Kembhavi (2023), which aim to integrate the exceptional pattern recognition strengths of deep neural networks with the inherent logical rigor and interpretability of symbolic systems. (2) Supervised Fine-Tuning (SFT) of MLLMs Xu et al. (2024a); Thawakar et al. (2025a), which relies on large-scale annotated datasets for end-to-end training to directly optimize model performance on specific visual reasoning tasks. (3) Reinforcement Learning (RL) based methods Tan et al. (2025); Huang et al. (2025), exemplified by techniques (e.g., Group Relative Policy Optimization (GRPO) Shao et al. (2024)). Such methods leverage RL's reward mechanisms to activate the latent reasoning potential within pretrained models, demonstrating favorable generalization capability, particularly when tackling complex visual-cognitive tasks involving mathematical logic derivation or code understanding, thus garnering increasing attention.

---

[1] In this work, we define **"action"** as a complete response (i.e., a full rollout), consisting of multiple generated tokens, rather than a single token prediction. This definition aligns with group-based rollout evaluation and is consistent with prior work such as GRPO Shao et al. (2024).

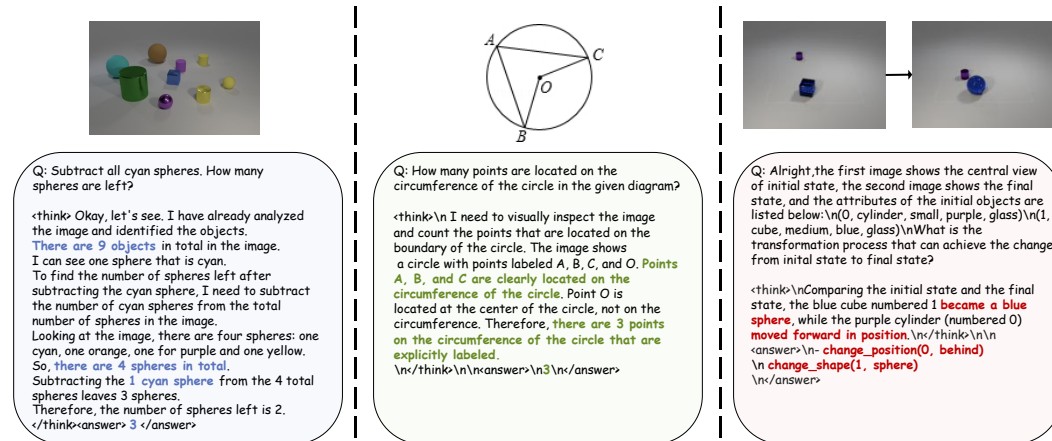

Figure 1: The output examples of Vision-EKIPL on three visual reasoning tasks, which shows superior reasoning results.

However, despite the notable successes achieved by RL-based methods on a series of visual reasoning tasks, recent studies Yue et al. (2025) have revealed a noteworthy phenomenon: the reasoning paths generated by models post-RL training appear, to a large extent, not to surpass the inherent capability scope of the pretrained foundation model. This suggests that the performance enhancements conferred by RL training might predominantly stem from its role as a preference optimizer. Specifically, RL reinforces the model's sampling strategy via reward signals, biasing it towards selecting known reasoning paths that have historically yielded high rewards, thereby more efficiently generating correct answers. Yet, this mechanism carries an inherent potential bottleneck: it may excessively favor the exploitation of known successful paths, consequently inhibiting the exploration of novel or more complex reasoning paths. A potential consequence is that the reasoning boundary of a RL-fine-tuned model, compared to its foundation model counterpart with vast potential, might not only fail to expand but could potentially constrict. Furthermore, existing RL methods commonly suffer from slow convergence rates and low training efficiency.

To overcome the dual limitations of current RL methods concerning reasoning boundary expansion and training efficiency, this paper introduces a novel reinforcement learning framework named Vision-EKIPL. Its core innovation lies in significantly broadening the sources of information during the policy learning process. At each input state, the framework not only *samples actions based on the current policy model but also incorporates actions from multiple external auxiliary models* into the candidate set. Subsequently, these candidate actions are ranked based on the reward signals they receive, and the top-$k$ highest-reward actions are selected to form a high-quality action group. The group is then utilized to guide the optimization of the policy model. Through this mechanism, Vision-EKIPL effectively broadens the policy model's exploration space by integrating potential solutions offered by diverse "experts" (i.e., the auxiliary models), aiding in the discovery of effective reasoning paths that might be overlooked by a single policy model.

In essence, our method aims to significantly elevate the reasoning frontier of the policy model by proactively introducing and integrating external knowledge (manifested as high-quality actions from auxiliary models) during the optimization process, enabling it to explore and learn richer, more complex reasoning strategies, thereby effectively mitigating the potential reasoning capacity attrition associated with standard reinforcement learning fine-tuning. Concurrently, by directly leveraging high-quality actions from external models to guide the optimization of the policy model, our method also substantially enhances the convergence rate and overall efficiency of the training process. From a broader perspective, this approach can be conceptualized as a promising hybrid paradigm of supervised fine-tuning (data distillation) and reinforcement learning. When the initial reasoning capacity of the policy model is comparatively limited, the model exhibits a greater propensity to select knowledge acquired from external models for supervised learning; conversely, as the policy model's own capability progressively advances, it gradually leans towards autonomous exploration of deeper reasoning strategies. Vision-EKIPL achieves up to a 5% performance improvement compared to the state-of-the-art(SOTA) on the Reason-RFT-CoT Benchmark. The output examples of Vision-EKIPL are provided in Fig. 1. Although this research primarily validates the efficacy of the framework on

visual reasoning tasks, the proposed framework possesses commendable generality and can theoretically be flexibly applied to a broader spectrum of artificial intelligence domains, including various linguistic tasks, visual tasks, and multimodal tasks.

Our main contributions can be summarized as follows:

- We propose Vision-EKIPL, an innovative reinforcement learning framework that significantly enhances the visual reasoning capability of MLLMs by integrating high-quality actions generated by external models to assist the optimization of the policy model.

- We demonstrate that incorporating high-quality actions from external models during policy optimization effectively broadens the policy model's action exploration space, thereby expanding its reasoning boundary.

- Through extensive experiment evaluation, we verify the effectiveness of the Vision-EKIPL framework, offering valuable insights for advancing visual reasoning research and introducing a new paradigm potentially conducive to promoting multimodal learning research.

## 2 METHOD

### 2.1 PRELIMINARIES

**Problem Definition.**  Visual reasoningAmizadeh et al. (2020a); Thawakar et al. (2025b); Xu et al. (2024b) can be formally defined as the task of inferring conclusions or answers by jointly analyzing visual and textual information. Given a visual input $I$ (e.g., images or videos) and an associated textual description or question $T$, the objective is to generate a corresponding answer $A$. This process can be formalized as:

$$P : (I, T) \rightarrow A$$

where $I \in \mathbb{R}^{H \times W \times C}$ denotes the visual input, characterized by height $H$, width $W$, and the number of channels $C$. The textual input $T$ typically consists of natural language queries or descriptions, while the output $A$ represents the inferred answer, which may be expressed in natural language or structured formats. Through this mapping, visual reasoning models are designed to effectively integrate and interpret multimodal information to perform complex reasoning tasks.

**Group Relative Policy Optimization (GRPO).**  GRPOShao et al. (2024) presents a novel reinforcement learning framework that has demonstrated strong performance in models such as DeepSeek R1Guo et al. (2025a). The fundamental objective of GRPO is to enhance the reasoning capability of model by iteratively refining its policy based on the relative performance of sampled actions within a group.

The process commences with the current policy $\pi_\theta$ for a given state $s$. A group of $N$ actions, $\{o_1, o_2, \ldots, o_N\}$, is sampled from the policy's output distribution $\pi_\theta(o|s)$. Each sampled action $o_i$ in this group is subsequently evaluated using a reward function $R(o_i)$, which quantifies the desirability or effectiveness of the action.

A key element of GRPO is the computation of an advantage score for each action. The advantage $A_i$ for the action $o_i$ is defined as:

$$A_i = \frac{R(o_i) - \text{mean}(\{R(o_1), R(o_2), \ldots, R(o_N)\})}{\text{std}(\{R(o_1), R(o_2), \ldots, R(o_N)\})} \quad (1)$$

Actions yielding a positive advantage are considered superior to the group average, while those with a negative advantage are deemed inferior. After computing the advantage $A_i$, GRPO evaluates the ratio of the probabilities of each action under the updated policy $\pi_{\theta_{\text{new}}}$ and the previous policy $\pi_{\theta_{\text{old}}}$, denoted as $\text{ratio}_i$.

$$\text{ratio}_i = \pi_{\theta_{\text{new}}}(o_i \mid s) \, / \, \pi_{\theta_{\text{old}}}(o_i \mid s) \quad (2)$$

The policy model parameters $\theta$ are then updated to increase the likelihood of selecting actions that demonstrated positive advantages and decrease the probability of choosing actions with negative advantages. This update is typically performed using gradient-based optimization method. To mitigate excessive policy updates and enhance training stability, GRPO constrains $\text{ratio}_i$ to the interval

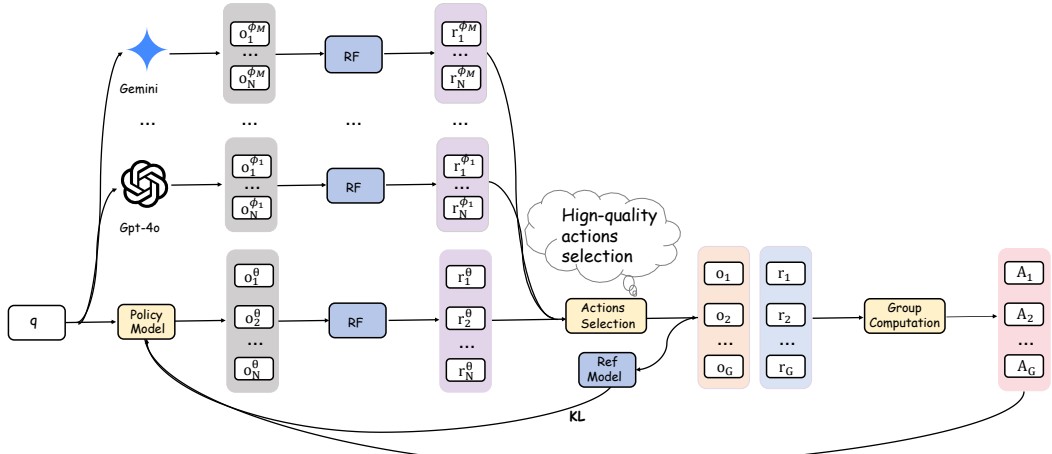

Figure 2: Overview of the proposed Vision-EKIPL framework. Vision-EKIPL samples high-quality action groups from the action sets of the external models and the policy model based on reward function (RF) evaluation, and then optimizes the policy model using the high-quality action group through the GRPO algorithm.

$[1 - \delta, 1 + \delta]$. Moreover, to encourage the learned policy to remain in proximity to the reference distribution $\pi_{\text{ref}}$, a Kullback-Leibler (KL) divergence penalty, weighted by a coefficient $\beta$, is integrated into the optimization objective. Finally, the optimization objective of GRPO can be formulated as follows:

$$
\begin{aligned}
\mathcal{J}_{GRPO}(\theta) = \mathbb{E}_{s \sim Q, \{o_i\}_{i=1}^N \sim \pi_{old}} & \left[ \frac{1}{N} \sum_{i=1}^N \min\left( \text{ratio}_i A_i, \right. \right. \\
& \left. \left. \text{clip}\left( \text{ratio}_i, 1 - \epsilon, 1 + \epsilon \right) A_i \right) - \beta \mathbb{D}_{KL}\left[ \pi_\theta || \pi_{ref} \right] \right]
\end{aligned}
\tag{3}
$$

where $Q$ denotes the candidate question set, $\mathbb{D}_{\text{KL}}$ denotes the KL regularization. $\pi_{\text{ref}}$ is typically a frozen pre-trained MLLM. In a nutshell, GRPO aims to maximize the expected advantage, often incorporating this KL divergence as a penalty term.

## 2.2 EXTERNAL KNOWLEDGE-INFUSED POLICY LEARNING

The overall framework of Vision-EKIPL is illustrated in Fig. 2. Vision-EKIPL is a reinforcement learning framework designed to enhance the visual reasoning capability of MLLMs. The key insight of Vision-EKIPL is leveraging high-quality actions generated by external auxiliary models to guide the optimization of the policy model, thereby infusing novel reasoning knowledge and further expanding the model's reasoning capacity.

**Sampling Action Groups beyond policy model.** We introduce a total of $M$ auxiliary models to support the learning process. Although the auxiliary actions are not solely derived from $\pi_{\theta_{\text{old}}}$, as long as the set of actions drawn, for example, from previous policy iterations or a dedicated exploration space, collectively encompasses the entire trajectory space that the current target policy $\pi_{\theta_{\text{new}}}$ can generate, then employing these mixed actions in importance sampling remains theoretically valid, with unbiasedness guaranteed. We have proven it theoretically in the appendix A. Given an input state $s = (x, q)$, where $x$ denotes the visual encoding of the input image and $q$ represents the textual encoding of the question, GRPO first samples a group of actions $\{o_1^\theta, \ldots, o_N^\theta\}$ from the current policy $\pi_\theta$. Additionally, for each auxiliary model $\pi^{\phi_j}$, it samples a corresponding group of actions $\{o_1^{\phi_j}, \ldots, o_N^{\phi_j}\}$. The sampling process is as follows:

$$
o_i^\theta \sim \pi_\theta(o \mid x, q), \quad \text{for } i = 1, 2, \ldots, N
\tag{4}
$$

$$o_i^{\phi_j} \sim \pi^{\phi_j}(o \mid x, q), \quad \text{for } i = 1, 2, \ldots, N, \tag{5}$$
$$\text{for } j = 1, 2, \ldots, M$$

All these sampled actions are then combined into a total action group $O$:

$$O = \{o_i^\theta \mid i = 1, \ldots, N\} \ \cup \ \bigcup_{j=1}^{M} \{o_i^{\phi_j} \mid i = 1, \ldots, N\} \tag{6}$$

**Reward Calculation.** Each sampled action $o_i$ is assigned a reward $R(o_i)$ based on verifiable criteria. In the context of visual reasoning tasks, the reward function $R(o_i)$Tan et al. (2025) integrates two components: a format reward $R_{\text{format}}(o_i)$ and an accuracy reward $R_{\text{acc}}(o_i)$. The format reward enforces adherence to a structured response format, while the accuracy reward assesses the correctness of the output, thereby striking a balance between structured reasoning and factual accuracy. The reward function is formally defined as:

$$R(o_i) = R_{\text{format}}(o_i) + R_{\text{acc}}(o_i). \tag{7}$$

The reward calculation follows the criteria outlined below:

- If the response provides a correct final answer,the model receives an positive accuracy reward. Otherwise, the model receives 0 reward. For the specific definition of accuracy reward, please refer to Tan et al. (2025).
- If the response encloses its reasoning within `<think></think>` tags and its final answer within `<answer></answer>` tags, the model receives a format reward of +1. Otherwise, the model receives 0 reward.

**Action Selection and Advantage Computation.** The actions within the action group $O$ are sorted in descending order based on their reward values, and the top-$G$ actions are selected to form the group of high-quality action $T : \{o_1, o_2, \ldots, o_G\}$, along with their corresponding group of rewards $R : \{r_1, r_2, \ldots, a_G\}$.

The rewards within the sampled reward group $R$ are normalized to compute the relative advantages $\{A_1, A_2, \ldots, A_G\}$, which are computed as shown in Eqn. 1. After computing the relative advantages for the action group $T$, the policy model is updated following Eqn. 3.

## 3 EXPERIMENT

### 3.1 EXPERIMENTAL DETAILS

In this paper, we employ the Reason-RFT-CoT Dataset Tan et al. (2025) to evaluate our method. The experiments are organized into the following three task categories: (1) **Visual Counting** This task assesses multimodal reasoning by integrating linguistic, visual, and mathematical skills to solve arithmetic problems within 3D block-based scenes. (2) **Structure Perception** This visual reasoning task requires models to interpret structural information across various mathematical geometries, medical imaging, chart layouts, and architectural designs. (3) **Spatial Transformation** This spatial-visual reasoning task evaluates a model's ability to infer single-step or multi-step transformation actions by analyzing initial and final visual states of 3D scenes presented from multiple perspectives (e.g., center, left, right). Each task contains in-domain test set and out-of-domain test set. Specific information can be found in Tan et al. (2025).

**Implementation Details** In our experiments, we utilize Qwen2-VL-2B and Qwen2-VL-7B Wang et al. (2024) as policy models. For external models, we selected GPT-4o Hurst et al. (2024) and Gemini-1.5-Pro Team et al. (2024). Our implementation is based on the open-source frameworks Open-R1 Huggingface (2025) and vLLM Kwon et al. (2023) to ensure reproducibility of results and system scalability. For hyperparameters, we employed a cosine learning rate schedule with a peak value of $5 \times 10^{-7}$ and adopted the AdamW optimizer to optimize the policy model, setting N and G to 8, with a default KL penalty of $\beta = 0.005$. Based on the dataset size, we set the number of epochs for Visual Counting task, Spatial Transformation task, and Structure Perception task to 1, 1, and 5, respectively. To ensure stability and statistical significance of the results, we repeat each

major experiment three times under the same setting and report the average accuracy across runs as the final result. All experiments are conducted on a server equipped with 8 A100 GPUs.

**API Cost Considerations** Incorporating high-quality actions generated by external models such as GPT-4o and Gemini-1.5-Pro introduces non-negligible API costs. These costs include both inference latency (typically 1–3 seconds per query depending on sequence length and model load) and monetary expenses (approximately 0.01–0.03 per query, varying with provider pricing tiers). Although these costs are relatively minor in small-scale experiments, they may accumulate significantly in large-scale training or deployment scenarios. In future work, exploring cost-effective open-source alternatives or distillation-based approaches to mitigate reliance on expensive APIs could be a promising direction.

**Baselines for comparison** To evaluate the performance and generalization capabilities of different training strategies, adhering to the settings in Tan et al. (2025), the methods compared in this paper are as follows: (1) SFT-based methods—ANS-SFT, which fine-tunes on answer generation, and CoT-SFT, which uses supervised learning with chain-of-thought (CoT) reasoning. (2) RL-based methods——Reason-RFT-Zero, which applies RL training directly to the base model, Reason-RFT, which first performs supervised learning with partial chain-of-thought (CoT) data before RL training and Vision-EKIPL, which integrates the external auxiliary models into the RL training.

To conduct the comprehensive evaluation, we adopt Qwen2-VL-Instruct Wang et al. (2024) as the base model, assessing both its 2B and 7B variants to investigate the influence of model scale. Additionally, the most advanced open-source models Bai et al. (2025); Abdin et al. (2024); Li et al. (2024b); Chen et al. (2024); Meta AI (2024); Agrawal et al. (2024) and proprietary models Hurst et al. (2024); Team et al. (2024) are incorporated as baselines to assess the performance of various training paradigms.

Table 1: **Results on three visual reasoning tasks.** The best results among different training paradigms are highlighted in **bold**, while the second-best results are underlined. "ID" denotes in-domain test data, and "OOD" denotes out-of-domain test data.

| Method | Visual Counting | | | Structure Perception | | | Spatial Transformation | | | |
|---|---|---|---|---|---|---|---|---|---|---|
| | Clevr-Math ID | Super-Clevr OOD | AVG | GeoMath ID | Geometry3k OOD | AVG | TRANCE ID | TRANCE-L OOD | TRANCE-R | AVG |
| **Proprietary Models** | | | | | | | | | | |
| GPT-4o-2024-08-06 Hurst et al. (2024) | 68.10 | 34.31 | 51.20 | 50.18 | 43.49 | 46.83 | 42.55 | 28.67 | 29.76 | 35.88 |
| Gemini-1.5-Pro Team et al. (2024) | 61.80 | 37.50 | 49.65 | 50.12 | 48.38 | 49.45 | 26.22 | 18.76 | 19.88 | 22.77 |
| **Open-Source Models** | | | | | | | | | | |
| Qwen2.5-VL-3B-Instruct Bai et al. (2025) | 75.90 | 39.30 | 57.60 | 36.75 | 37.44 | 37.09 | 8.57 | 8.26 | 8.31 | 8.42 |
| Phi-3.5-Vision-4B-Instruct Abdin et al. (2024) | 21.40 | 15.20 | 18.30 | 36.83 | 50.25 | 43.54 | 7.42 | 2.45 | 4.02 | 5.33 |
| Llava-OneVision-7B Li et al. (2024b) | 69.70 | 29.10 | 49.40 | 77.63 | 43.66 | 60.64 | 10.00 | 8.33 | 8.74 | 9.27 |
| Qwen2.5-VL-7B-Instruct Bai et al. (2025) | 74.60 | 35.20 | 54.90 | 44.00 | 45.61 | 44.80 | 19.63 | 13.12 | 13.42 | 16.45 |
| InternVL-2.5-8B Chen et al. (2024) | 93.50 | 35.30 | 64.40 | 63.00 | 47.32 | 51.60 | 7.19 | 6.62 | 6.63 | 6.91 |
| Llama-3.2-11B-Vision Meta AI (2024) | 10.30 | 9.50 | 9.90 | 13.75 | 20.85 | 17.30 | 8.22 | 8.40 | 9.03 | 8.47 |
| Pixtral-12B Agrawal et al. (2024) | 42.60 | 22.90 | 32.75 | 30.38 | 36.09 | 33.23 | 7.35 | 5.03 | 5.22 | 6.42 |
| **Qwen2VL-2B-Instruct** | | | | | | | | | | |
| Zero-Shot | 82.40 | 32.00 | 57.20 | 25.86 | 20.63 | 23.25 | 3.78 | 4.60 | 4.67 | 4.35 |
| + ANS-SFTTan et al. (2025) | 96.20 | 39.20 | 67.70 | **51.34** | 22.50 | 36.92 | 77.39 | 49.24 | 50.33 | 58.99 |
| + CoT-SFTTan et al. (2025) | 85.50 | 46.50 | 66.00 | 43.05 | 25.25 | 34.15 | 64.37 | 43.19 | 42.86 | 50.14 |
| + Reason-RFT-ZeroTan et al. (2025) | 98.40 | 44.80 | 71.60 | 47.68 | 32.50 | 40.09 | 42.13 | 34.07 | 33.41 | 33.74 |
| + Reason-RFTTan et al. (2025) | 96.80 | 51.20 | 74.00 | 49.03 | 33.13 | 41.08 | 74.61 | 64.05 | 64.08 | 67.58 |
| + Vision-EKIPL(Ours) | **99.10** | **52.30** | **75.70** | 49.70 | **34.50** | **42.10** | **78.23** | **65.12** | **65.45** | **69.60** |
| **Qwen2VL-7B-Instruct** | | | | | | | | | | |
| Zero-Shot | 98.60 | 42.10 | 70.35 | 43.30 | 43.88 | 43.59 | 13.53 | 12.72 | 12.78 | 13.01 |
| + ANS-SFTTan et al. (2025) | 95.00 | 33.90 | 64.45 | 51.34 | 25.38 | 38.36 | 82.19 | 54.29 | 54.83 | 63.77 |
| + CoT-SFTTan et al. (2025) | 87.30 | 42.40 | 64.85 | 50.49 | 33.00 | 41.75 | 81.31 | 47.90 | 47.80 | 59.00 |
| + Reason-RFT-ZeroTan et al. (2025) | 99.40 | 53.00 | 76.20 | 55.00 | 54.75 | 54.88 | 67.67 | 57.20 | 56.15 | 56.68 |
| + Reason-RFTTan et al. (2025) | 95.60 | 51.00 | 73.30 | 59.27 | 49.25 | 54.26 | 79.97 | 59.36 | 58.61 | 65.98 |
| + Vision-EKIPL(Ours) | **99.70** | **53.30** | **76.50** | **60.10** | **56.75** | **58.42** | **83.32** | **62.35** | **60.47** | **68.71** |

## 3.2 MAIN RESULTS

**Results on In-Domain Tasks** To evaluate the In-Domain (ID) performance of Vision-EKIPL relative to different training paradigms and baseline models across visual reasoning tasks, we conducted extensive training and evaluation on 2B/7B models for three tasks. The results, presented in Tab. 1, indicate the following: (1) **Visual Counting** RL-based methods consistently outperformed all

open-source and proprietary baseline models, as well as SFT-based methods, across both 2B and 7B models, with Vision-EKIPL achieving the best performance among the 7B models; (2) **Structure Perception** RL-based methods surpassed SFT-based methods in the 7B model, while ANS-SFT demonstrated the best performance in the 2B model. CoT-SFT showed limited improvement, potentially because enforced reasoning supervision hindered cognitive enhancement. Furthermore, Vision-EKIPL in the 7B model outperformed all proprietary models and most open-source models, with the exception of InternVL-2.5-8B Chen et al. (2024) and Llava-OneVision-7B Li et al. (2024a); (3) **Spatial Transformation** Vision-EKIPL achieves the highest performance, surpassing all baseline models. Unlike Reason-RFT, Vision-EKIPL does not require supervised fine-tuning to activate its reasoning capacity, yet still outperforms Reason-RFT. This demonstrates that incorporating high-quality actions from external models can effectively raise the model's reasoning capacity.

**Results on Out-of-Domain Generalization** To validate the out-of-domain (OOD) performance of Vision-EKIPL relative to different training paradigms and baseline models across visual reasoning tasks, we conducted comprehensive experiments on 2B/7B models for three tasks. The results, presented in Tab. 1, reveal the following: (1) **Visual Counting** RL-based methods demonstrate superior generalization capability compared to SFT-based methods in both 2B and 7B models. Specifically, Vision-EKIPL outperforms ANS-SFT by 13% (2B) and 19% (7B), and also surpasses all open-source and proprietary baselines. Notably, compared to traditional RL methods (e.g., Reason-RFT), Vision-EKIPL significantly expands the model's reasoning boundary, enabling the model to explore and find correct reasoning paths on complex problems that Reason-RFT could not find. (2) **Structure Perception** RL-based methods consistently outperform SFT-based methods, with Vision-EKIPL achieving the best results in both 2B and 7B models (8% higher than Reason-RFT on 2B model), while Reason-RFT achieves comparable performance in the 2B model. SFT-based methods shows limited impact, especially in the 7B model; (3) **Spatial Transformation** RL-based methods surpass SFT-based methods in both 2B and 7B models, while significantly outperforming all baseline models. Vision-EKIPL (2B) exhibits exceptional OOD generalization capability, exceeding GPT-4o Hurst et al. (2024) by 34% and Gemini-1.5-Pro Team et al. (2024) by 47%. Overall, Vision-EKIPL surpasses all open-source and proprietary baselines, as well as other training methods, demonstrating exceptional performance in visual reasoning generalization capability.

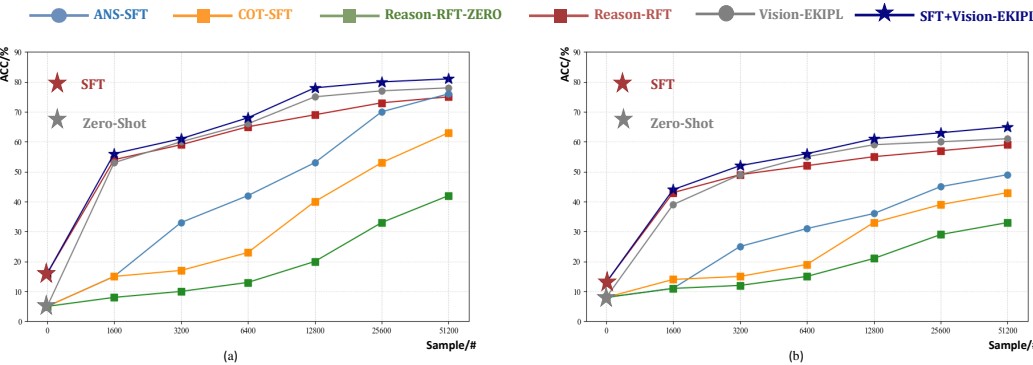

Figure 3: Results of different methods on the Spatial Transformation task across training processes. (a) Evaluation results for 2B model on ID task, (b) Evaluation results for 2B model on OOD task.

### 3.3 TRAINING EFFICIENCY EVALUATION

To demonstrate the data efficiency of Vision-EKIPL during training, we trained all methods on the TRANCE dataset and recorded intermediate and validation results, as illustrated in Fig. 3. Vision-EKIPL demonstrates excellent data efficiency in both in-domain (ID) and out-of-domain (OOD) tasks. The main findings include: (1) On ID tasks, Vision-EKIPL surpasses the performance of Reason-RFT using only 25% of the training data (12,800 samples). Furthermore, when Vision-EKIPL underwent SFT training using the CoT dataset before RL training, following the settings in Tan et al. (2025), it achieves 93% of Reason-RFT's performance using only 12% of the training data. (2) On OOD tasks, Vision-EKIPL achieves the performance of Reason-RFT using only 12% of the data, demonstrating strong generalization capability.

## 3.4 ANALYSIS ON THE SOURCES OF ACTION

As illustrated in Fig.4, we tracked the dynamic evolution of the ratio of actions sampled from the external models and the policy model within the group of actions utilized for parameter updates during the training process of the 2B model on the TRANCE dataset. We can observe that, as the

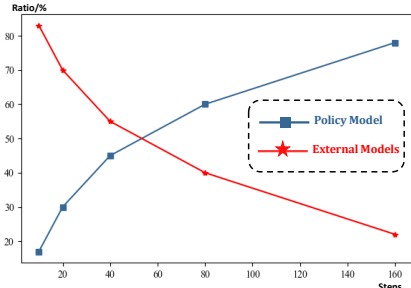

Figure 4: Ratio of actions sampled from the external models and the policy model

number of training iterations increases, the proportion of actions originating from the external models gradually decreases, while the proportion of actions from the policy model itself progressively increases within the group of actions used for updating the policy model's parameters.

This phenomenon can be attributed to the initial stages of training: the reasoning capability of the policy model is relatively weaker compared to the auxiliary models (or external models). Consequently, actions sampled from the policy model typically receive lower rewards than those provided by the auxiliary models. To effectively guide the optimization direction of the policy model, we prioritize the selection of actions generated by the auxiliary models.

However, with further model optimization and deeper training, the reasoning capability of the policy model improves significantly, and the rewards obtained from its generated actions also increase accordingly. At this stage, to fully leverage the policy model's own learning outcomes and accelerate its convergence, we increasingly select actions generated by the policy model to drive the model's optimization.

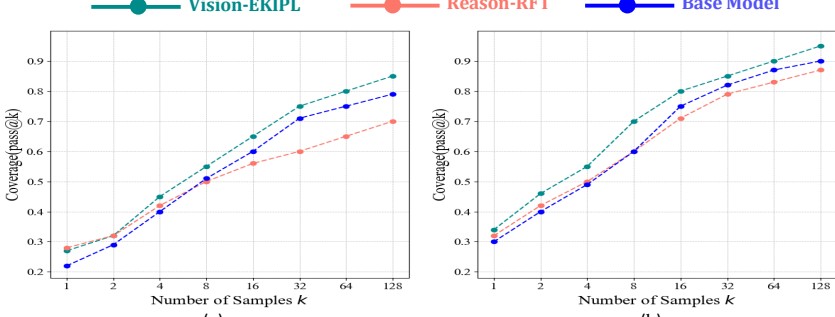

Figure 5: Pass@k curves of base model,Reason-RFT and Vision-EKIPL on the Spatial Transformation task. Evaluation results for 2B model (a) and 7B model (b) on ID task.

## 3.5 PUSH FORWARD THE BOUNDARY OF THE MODEL'S REASONING ABILITY

As shown in Fig. 5, we conduct a comparative analysis of the Pass@$K$ scores for Vision-EKIPL, Reason-RFT, and the base model under varying $k$ values. The Pass@$K$ metric reflects the likelihood that a model produces at least one correct answer across $K$ independent samples, and thus serves as a proxy for evaluating the upper bound of the model's reasoning capability when given sufficient exploration opportunities.

At lower values of $k$, both Vision-EKIPL and Reason-RFT outperform the base model, indicating that reinforcement learning helps guide the model toward more accurate reasoning paths in the early stages. However, a noteworthy phenomenon emerges as $k$ increases: the Pass@$K$ score of Reason-RFT begins to fall behind that of the base model. This suggests that although Reason-RFT improves

sample efficiency, it may limit the diversity of the model's reasoning space by overemphasizing high-reward but familiar reasoning patterns, thus constraining its exploratory capacity.

In contrast, Vision-EKIPL demonstrates a consistently superior performance across all $k$ values. Especially at larger $k$, its Pass@$K$ score significantly surpasses both the base model and Reason-RFT. This clearly highlights Vision-EKIPL's ability to push the reasoning frontier forward. Its core advantage lies in the integration of high-quality actions generated by external expert models during training. These actions serve as distilled supervision targets, enabling the policy model to learn diverse and novel reasoning strategies that would otherwise be inaccessible through self-sampling.

## 4 RELATED WORK

**Visual Reasoning**  This technology has broad application prospects, including visual counting Lindström & Abraham (2022); Li et al. (2023), geometric problem solving Gao et al. (2023); Kazemi et al. (2023); Lu et al. (2023); Zhang et al. (2024b); Shi et al. (2024), visual transformation reasoning Hong et al. (2021), scientific research Lu et al. (2022); Kembhavi et al. (2016), and robotic task planning Hu et al. (2023); Ji et al. (2025); Hao et al. (2025a). Early work in visual reasoning relied on programmatic generation Johnson et al. (2017); Gupta & Kembhavi (2023); Surís et al. (2023) or neuro-symbolic methods Garcez et al. (2019); Amizadeh et al. (2020b); Choi et al. (2024); Zhang et al. (2024a). In recent years, driven by the rapid development of MLLMs, the field has seen breakthrough progress. For example, LLaVA-CoT Xu et al. (2024a) employs a multi-stage Chain-of-Thought (CoT) Wei et al. (2022) supervised fine-tuning (SFT) strategy, while Insight-V Dong et al. (2024) combines SFT with reinforcement learning (RL). DeepSeek-R1-Zero Guo et al. (2025a) introduced a rule-based RL approach, significantly enhancing reasoning capability. Building upon DeepSeek-R1 Guo et al. (2025a), we propose a novel RL method that substantially improves the model's reasoning performance.

**Reinforcement Learning**  Reinforcement learning (RL) has demonstrated significant efficacy in enhancing the reasoning capabilities of Large Language Models (LLMs) through iterative, feedback-driven refinement Christiano et al. (2017); Silver et al. (2017); Shao et al. (2024); Yang et al. (2024a); Ying et al. (2024); Hui et al. (2024); Zhang et al. (2024c). Notable methodologies include Reinforcement Learning from Human Feedback (RLHF) Ouyang et al. (2022) and Reinforcement Learning from AI Feedback (RLAIF) Bai et al. (2022), both of which leverage either human or AI-generated feedback to refine model behavior. Within the domain of vision-language tasks, RL has been successfully employed to align model predictions with human preferences and mitigate the occurrence of hallucinations Sun et al. (2023); Yu et al. (2024a;b); Zhao et al. (2023). More recently, advancements such as DeepSeek-R1-Zero Guo et al. (2025b) have introduced GRPO Shao et al. (2024), a technique that utilizes rule-based rewards to strengthen reasoning abilities without requiring supervised fine-tuning. GRPO has been further adapted for specialized applications, with Visual-RFT Liu et al. (2025) employing it for visual grounding and Med-R1 Pan et al. (2025) applying it to medical reasoning tasks. Vision-R1 Huang et al. (2025) and Reason-RFT Tan et al. (2025) adopt a two-stage training paradigm—CoT supervised fine-tuning followed by GRPO-based reinforcement fine-tuning—to enhance the reasoning performance of MLLMs. Distinctly, our Vision-EKIPL is the first to leverage high-quality actions generated by external models to guide policy-model optimization, thereby infusing novel reasoning knowledge and significantly advancing reasoning ability.

## 5 CONCLUSION

In this paper, we propose Vision-EKIPL, a novel reinforcement learning framework designed to enhance the generalization capability of visual reasoning models. This is achieved by skillfully introducing high-quality actions generated by external auxiliary models to guide the optimization of the policy model during the RL training process. This innovative approach significantly expands the model's exploration space, enabling the model to effectively surpass traditional reasoning boundary, while also substantially accelerating training convergence speed and overall efficiency. Extensive experiments demonstrate the effectiveness of Vision-EKIPL, providing valuable insights for advancing visual reasoning research and introducing a new paradigm in multimodal learning.

## 6 ETHICS STATEMENT

This work introduces a self-rewarding direct preference optimization framework aimed at improving the reasoning reliability of Large Vision-Language Models (LVLMs). All experiments are conducted using publicly available datasets and open-source base models, ensuring that no private or sensitive data is involved. We acknowledge that, despite the integration of process-level reward modeling and memory-based error avoidance mechanisms, the model may still produce incorrect or biased outputs. Therefore, we strongly discourage the use of our method in high-stakes or ethically sensitive scenarios, such as generating deceptive information or automating critical decision-making processes. In accordance with ICLR's Code of Ethics, all contributions have been properly disclosed, and the authors assume full responsibility for the content and claims made in this paper.

## 7 REPRODUCIBILITY STATEMENT

This study is committed to ensuring that the experimental results of our method are fully reproducible. All experiments were conducted on a server equipped with 8 A100 GPUs.

Our implementation is based on two open-source frameworks: Open-R1 and vLLM. The code will be made publicly available upon the final publication of the paper to facilitate reproducibility within the research community.

The following models were used in our experiments:

- **Policy Models:** Qwen2-VL-2B and Qwen2-VL-7B.
- **External Auxiliary Models:** GPT-4o and Gemini-1.5-Pro.

The experimental evaluation utilized the Reason-RFT-COT dataset, which contains three types of tasks:

- **Visual Counting:** Assesses multi-modal reasoning capabilities to solve arithmetic problems based on 3D block scenes.
- **Structure Perception:** Requires the model to interpret structural information in various mathematical geometries, medical images, chart layouts, and architectural designs.
- **Spatial Transformation:** Evaluates the model's ability to analyze the initial and final visual states of a 3D scene and infer single-step or multi-step transformation actions.

The main hyperparameter settings were as follows: We adopted a cosine learning rate schedule with a peak value of $5 \times 10^{-7}$ and used the AdamW optimizer to optimize the policy model. The hyperparameters $N$ and $G$ were both set to 8, and the default KL penalty was $\beta = 0.005$. To ensure the stability and statistical significance of our results, all main experiments were repeated three times under the same settings, and the average accuracy was reported as the final result. Based on the dataset size, the number of training epochs for the visual counting, spatial transformation, and structure perception tasks were set to 1, 1, and 5, respectively.

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

## A  IMPORTANCE SAMPLING CORRECTION AND ITS PROPERTIES

This section presents a rigorous derivation and properties of importance sampling when samples are drawn from a general proposal distribution $q(o \mid s)$, rather than the reference policy $\pi_{\theta_{\text{old}}}$. We introduce the unbiased IS transformation, the self-normalized IS (SNIS) estimator, bias bounds, the effective sample size (ESS), and explicit formulas for mixture proposals.

### A.1  BASIC IDENTITY AND CORRECTION

Consider the expectation under the reference policy:

$$J \;=\; \mathbb{E}_{o \sim \pi_{\theta_{\text{old}}}(\cdot \mid s)}\big[g(o)\big],$$

where $g(o)$ denotes any integrable function (in our case, often the product of a policy ratio and an advantage term, e.g., $g(o) = \frac{\pi_{\theta_{\text{new}}}(o \mid s)}{\pi_{\theta_{\text{old}}}(o \mid s)} A(o)$). When samples are drawn from a proposal $q(o \mid s)$, the unbiased IS transformation is obtained via the importance weight

$$w(o) \;=\; \frac{\pi_{\theta_{\text{old}}}(o \mid s)}{q(o \mid s)},$$

yielding

$$J = \mathbb{E}_{o \sim \pi_{\theta_{\text{old}}}}\big[g(o)\big] = \mathbb{E}_{o \sim q}\left[\frac{\pi_{\theta_{\text{old}}}(o \mid s)}{q(o \mid s)}\, g(o)\right] = \mathbb{E}_{o \sim q}\big[w(o)\, g(o)\big]. \tag{8}$$

For $g(o) = \frac{\pi_{\theta_{\text{new}}}(o \mid s)}{\pi_{\theta_{\text{old}}}(o \mid s)} A(o)$, Eq. equation 8 reduces to

$$J = \mathbb{E}_{o \sim q}\left[\frac{\pi_{\theta_{\text{old}}}(o \mid s)}{q(o \mid s)} \cdot \frac{\pi_{\theta_{\text{new}}}(o \mid s)}{\pi_{\theta_{\text{old}}}(o \mid s)} A(o)\right] = \mathbb{E}_{o \sim q}\left[\frac{\pi_{\theta_{\text{new}}}(o \mid s)}{q(o \mid s)} A(o)\right]. \tag{9}$$

Hence, when using samples from a proposal $q$, the correct unbiased estimator requires the ratio $\frac{\pi_{\theta_{\text{new}}}(o)}{q(o)}$, rather than $\frac{\pi_{\theta_{\text{new}}}(o)}{\pi_{\theta_{\text{old}}}(o)}$ (unless $q = \pi_{\theta_{\text{old}}}$).

### A.2  SAMPLE-BASED ESTIMATORS

Suppose we draw $o_1, \ldots, o_n \sim q$, with weights $w_i = w(o_i)$ and function values $g_i = g(o_i)$.

**Unnormalized IS:**

$$\widehat{J}_{\text{IS}} \;=\; \frac{1}{n} \sum_{i=1}^{n} w_i\, g_i,$$

which is unbiased under mild regularity conditions.

**Self-Normalized IS (commonly used in practice):**

$$\widehat{J}_{\text{SNIS}} \;=\; \sum_{i=1}^{n} \tilde{w}_i\, g_i, \qquad \tilde{w}_i \;=\; \frac{w_i}{\sum_{j=1}^{n} w_j}.$$

This reduces variance but introduces a small bias that vanishes as $n \to \infty$.

### A.3  EFFECTIVE SAMPLE SIZE (ESS)

The dispersion of normalized weights can be measured by the effective sample size:

$$\text{ESS} \approx \frac{\left(\sum_{i=1}^{n} w_i\right)^2}{\sum_{i=1}^{n} w_i^2}.$$

When $q$ is close to $\pi_{\theta_{\text{old}}}$, the weights are nearly constant and $\text{ESS} \approx n$; if weights are highly imbalanced, ESS becomes much smaller, leading to unstable estimates. In practice, monitoring ESS, weight variance, or high quantiles (e.g., 95% percentile of weights) provides insight into IS quality.

### A.4 BIAS BOUND VIA TOTAL VARIATION

If one ignores the correction and directly uses $\mathbb{E}_q[g]$ as an approximation to $J$, the bias can be bounded as

$$\left| \mathbb{E}_{\pi_{\theta_{\mathrm{old}}}}[g] - \mathbb{E}_q[g] \right| = \left| \int g(o)\big(\pi_{\theta_{\mathrm{old}}}(o) - q(o)\big)\, \mathrm{d}o \right| \le \|g\|_\infty \int \big|\pi_{\theta_{\mathrm{old}}}(o) - q(o)\big|\, \mathrm{d}o$$
$$= 2\|g\|_\infty \cdot \mathrm{TV}\big(\pi_{\theta_{\mathrm{old}}}, q\big), \tag{10}$$

where $\mathrm{TV}(p,q) = \frac{1}{2}\int |p-q|$ is the total variation distance. Thus, the approximation bias is jointly controlled by the function magnitude and the distributional divergence.

### A.5 EXPLICIT WEIGHTS FOR MIXTURE PROPOSALS

In many settings, the proposal $q$ is a mixture of multiple policies. For instance, if samples are pooled from the reference policy $\pi_{\theta_{\mathrm{old}}}$ and $M$ auxiliary policies $\{\pi_{\phi_j}\}_{j=1}^M$, the unfiltered proposal can be modeled as

$$q(o \mid s) \;=\; \frac{1}{M+1}\Big(\pi_{\theta_{\mathrm{old}}}(o \mid s) + \sum_{j=1}^M \pi_{\phi_j}(o \mid s)\Big),$$

with the importance weight

$$w(o) \;=\; \frac{\pi_{\theta_{\mathrm{old}}}(o \mid s)}{\frac{1}{M+1}\big(\pi_{\theta_{\mathrm{old}}}(o \mid s) + \sum_{j=1}^M \pi_{\phi_j}(o \mid s)\big)}.$$

If additional selection is performed (e.g., top-$G$ filtering), the actual proposal becomes the conditional distribution

$$q_{\mathrm{sel}}(o \mid s) \;=\; \frac{q(o \mid s)\, \mathbf{1}\{o \in S\}}{\int q(o \mid s)\, \mathbf{1}\{o \in S\}\, \mathrm{d}o},$$

where $S$ denotes the selection set. Corresponding weights are then computed as $\pi_{\theta_{\mathrm{old}}}(o \mid s)/q_{\mathrm{sel}}(o \mid s)$.

### A.6 ALGORITHMIC PSEUDOCODE

---

**Algorithm 1** Importance-Weighted Estimation with Proposal $q$

---

**Require:** Samples $\{o_i\}_{i=1}^n \sim q(o \mid s)$; function values $g_i = g(o_i)$; reference and candidate policy densities
**Ensure:** IS estimate $\widehat{J}_{\mathrm{IS}}$, SNIS estimate $\widehat{J}_{\mathrm{SNIS}}$, ESS

1: Compute weights $w_i \leftarrow \dfrac{\pi_{\theta_{\mathrm{old}}}(o_i \mid s)}{q(o_i \mid s)}$

2: $\widehat{J}_{\mathrm{IS}} \leftarrow \dfrac{1}{n}\sum_{i=1}^n w_i\, g_i$

3: Normalize: $\tilde{w}_i \leftarrow \dfrac{w_i}{\sum_{j=1}^n w_j}$

4: $\widehat{J}_{\mathrm{SNIS}} \leftarrow \sum_{i=1}^n \tilde{w}_i\, g_i$

5: $\mathrm{ESS} \leftarrow \dfrac{\big(\sum_{i=1}^n w_i\big)^2}{\sum_{i=1}^n w_i^2}$

6: **return** $\widehat{J}_{\mathrm{IS}},\ \widehat{J}_{\mathrm{SNIS}},\ \mathrm{ESS}$

---

### A.7 DISCUSSION AND PRACTICAL REMARKS

- If $q = \pi_{\theta_{\mathrm{old}}}$, then $w \equiv 1$, and Eq. equation 9 reduces to the standard policy ratio form $\frac{\pi_{\theta_{\mathrm{new}}}}{\pi_{\theta_{\mathrm{old}}}} A(o)$.

- When $q$ deviates substantially from $\pi_{\theta_{\mathrm{old}}}$, IS or SNIS correction is essential. Monitoring ESS and weight statistics provides diagnostics for stability.

- If $q$ is analytically intractable (e.g., after complex filtering), approximate calculations of $q(o)$ and corresponding bias analysis should be reported. Ablation comparing (A) uncorrected ratios, (B) exact IS, and (C) SNIS can empirically validate the approximation.

## A.8 SUMMARY

For samples drawn from a general proposal $q$, unbiased estimation of expectations under $\pi_{\theta_{\text{old}}}$ requires incorporating importance weights $w = \frac{\pi_{\theta_{\text{old}}}}{q}$. In the common case $g(o) = \frac{\pi_{\theta_{\text{new}}}}{\pi_{\theta_{\text{old}}}} A(o)$, this is equivalent to using the corrected ratio $\frac{\pi_{\theta_{\text{new}}}}{q}$. Approximate forms that drop $w$ may still be employed in practice but should be accompanied by theoretical justification (e.g., bias bound in Eq. equation 10) and empirical validation via ESS and ablation studies.

# B    LLM USAGE FOR PAPER WRITING

In the preparation of this paper, large language models (LLMs) were used solely for the purpose of polishing and refining the English language expression of the original author-written content. This includes improving grammatical fluency, sentence structure, and overall readability. All technical content, experimental results, methodological descriptions, figures, tables, and conclusions were generated and verified exclusively by the authors. The use of LLMs was strictly limited to post-writing language enhancement and did not involve any contribution to the scientific reasoning, analysis, or intellectual substance of the work. In accordance with ICLR's Policy 1 on LLM usage, we disclose this assistance here and in the submission form. Per ICLR's Policy 2, the authors take full responsibility for the entirety of the paper's content, including any language edits facilitated by LLMs.

