# OpenReview forum: "Vision-EKIPL: External Knowledge-Infused Policy Learning for Visual Reasoning"
_ICLR.cc/2026/Conference — Submitted to ICLR 2026_

### Official Review · Reviewer_vu6V · 2025-10-17

**Soundness:** 1
**Presentation:** 2
**Contribution:** 1
**Rating:** 0
**Confidence:** 4

**Summary:**

This paper introduces Vision-EKIPL, a reinforcement learning (RL) framework designed to improve the visual reasoning capabilities of Multimodal Large Language Models (MLLMs). The core idea is to augment the standard GRPO training process by incorporating "high-quality actions" (i.e., full-text responses) generated by powerful external models like GPT-4o and Gemini. These external responses are pooled with the policy model's own generations, ranked by a reward function, and the top-ranked responses are then used to update the policy model. The authors claim that this "external knowledge infusion" expands the model's reasoning boundary and accelerates training. Experiments are conducted on subsets of several niche academic benchmarks (Clevr-Math, Geometry3k, TRANCE), where the proposed method shows performance gains over a baseline RL method.

**Strengths:**

1. Clear Presentation: The paper is generally well-written and the proposed framework is explained in an easy-to-understand manner.

**Weaknesses:**

This paper suffers from a severe lack of novelty and is built upon a questionable experimental foundation, making its contributions insignificant and its conclusions unconvincing.
1. Fundamentally Unoriginal Method: The core contribution of this paper is, in essence, a form of knowledge distillation, a technique that is decades old and has been exhaustively explored in machine learning. The authors attempt to rebrand this by embedding it within a GRPO framework, but this does not mask the underlying simplicity: they are using stronger models to teach a weaker one. Labeling this as "External Knowledge-Infused Policy Learning" is a verbose attempt to dress up a well-worn concept. There is no meaningful technical or theoretical innovation here. The claim that this represents a new "paradigm" is a gross overstatement.
2. Highly Questionable and Insufficient Evaluation: The empirical evidence provided is deeply flawed and insufficient to support the paper's claims.
  - Cherry-Picked and Obscure Benchmarks: The authors evaluate their method on a narrow set of highly specific, synthetic, or academic datasets (Clevr-Math, Geometry3k, TRANCE). These are not representative of the broad, challenging reasoning tasks that the community is currently focused on. To make a convincing case, the method must be validated on widely recognized and challenging multimodal benchmarks such as MathVista, MathVerse, MMMU, and LogicVista. The current evaluation of niche datasets makes it impossible to gauge the true capability and generalizability of the model.
  - Lack of Meaningful Baselines: The primary comparison is against a previous method ("Reason-RFT"). There is a glaring lack of comparison against current state-of-the-art open-source models (e.g., the latest Qwen-VL models, InternVL 2.5) or, more importantly, the powerful proprietary models used as "teachers" (GPT-4o, Gemini). Showing that a student model improves by learning from a vastly superior teacher model is an entirely expected and uninteresting result. The crucial question is whether this complex and costly training scheme produces a model that can truly compete with top-tier models, and this has not been demonstrated.
3. Meaningless Contribution: The paper's conclusion—that incorporating guidance from superior models improves a weaker model's performance—is trivial. It offers no new scientific insight. The entire framework is predicated on having access to more powerful, expensive, closed-source models. This makes the method not only unoriginal but also impractical and of little value to the broader research community that may not have the resources to constantly query costly APIs for training.

**Questions:**

1. The core of your method is knowledge distillation. Could you please justify why this re-packaging within an RL framework should be considered a novel contribution to the field, rather than an application of a standard technique?
2. Your evaluation is confined to a very narrow set of academic datasets. To substantiate your claims of "significantly enhancing visual reasoning," I request that you evaluate your final 7B model on at least three of the following widely-used benchmarks: MathVista, MathVerse, MMMU, We-Math, MMKI2, or LogicVista.

---

### Official Review · Reviewer_3HTs · 2025-10-29

**Soundness:** 2
**Presentation:** 3
**Contribution:** 2
**Rating:** 4
**Confidence:** 4

**Summary:**

The paper introduces Vision-EKIPL, a reinforcement learning framework designed to enhance the visual reasoning ability of MLLMs. Unlike prior methods such as GRPO or Reason-RFT that rely solely on self-sampled actions, Vision-EKIPL incorporates high-quality actions generated by external expert models during training. These external actions are used to guide policy optimization, expanding the exploration space and mitigating over-exploitation of known reasoning paths. Experiments on the Reason-RFT-CoT benchmark show up to a 5% improvement over existing RL baselines across visual counting, structural perception, and spatial transformation tasks. The framework also claims faster convergence and better generalization, demonstrating potential as a hybrid between supervised distillation and RL for improving multimodal reasoning.

**Strengths:**

1. Proposes injecting external expert actions into GRPO training to expand exploration beyond on-policy rollouts—an intuitive hybrid of RLVR and distillation tailored to visual reasoning.

2. Clear algorithmic pipeline, reasonable reward design and multi-seed evaluation on Reason-RFT-CoT with 2B/7B backbones; reports up to ~5% gains over RL baselines and competitive OOD results.

3. Method is easy to follow, with an unbiasedness argument placed in the appendix; empirical curves help interpret behavior.

**Weaknesses:**

1. This work ignores closely related works. The core idea substantially overlaps with Learning to Reason under Off-Policy Guidance (LUFFY), which mixes off-policy traces with on-policy rollouts and introduces regularized importance sampling. The paper does not compare against LUFFY or discuss differences, which is critical to establish novelty and effectiveness.

2. Current evaluation is narrow. This paper should include stronger and recent benchmarks like MMMU-Pro, MathVisio to show real performance gains and OOD robustness.

3. The paper could provide a deeper literature review on slow-thinking reasoning MLLMs. To properly position the contribution and clarify what is novel, the paper can discuss the differences in objectives, training signals, and data cost:

[1]  Wei, Y. et al., "Open Vision Reasoner: Transferring Linguistic Cognitive Behavior for Visual Reasoning".

[2] Shen, J. et al., "Semi-off-Policy Reinforcement Learning for Vision-Language Slow-Thinking Reasoning".

[3] Chen, L. et al., "R1-V: Reinforcing Super Generalization Ability in Vision-Language Models with Less Than $3".

[4] Meng, F. et al., "MM-Eureka: Exploring Visual Aha Moment with Rule-based Large-scale Reinforcement Learning".

[5] Yao, H. et al., "R1-ShareVL: Incentivizing Reasoning Capability of Multimodal Large Language Models via Share-GRPO".

**Questions:**

See above weaknesses.

---

### Official Review · Reviewer_vVCs · 2025-11-01

**Soundness:** 2
**Presentation:** 3
**Contribution:** 2
**Rating:** 2
**Confidence:** 4

**Summary:**

This paper focuses on enhancing the visual reasoning capability of Multimodal Large Language Models. It proposes a novel RL framework named Vision-EKIPL to address the limitations of existing RL-based visual reasoning methods, such as constrained reasoning boundaries (actions sampled solely from the policy model itself), low training efficiency, and over-reliance on known reasoning paths. By integrating high-quality actions generated by more powerful external auxiliary models with those sampled by the policy model to form a candidate set, the framework guides the optimization of the policy model, thereby effectively expanding the model's exploration space, breaking through reasoning boundaries, and accelerating training convergence. Across multiple visual tasks, this framework achieves a performance improvement of up to 5% compared to the state-of-the-art methods.

**Strengths:**

The presentation of this paper is good and easy to follow.

**Weaknesses:**

- Insufficient innovation. Similar ideas have been fully discussed by Luffy [1]. The authors are advised to discuss the similarities and differences between this work and that paper.
- Overly strong assumption. As mentioned by the authors in Line 208, the premise for directly mixing internal and external model trajectories is that the trajectories from external models can be generated by the current policy model. However, this is an overly strong assumption, especially given the authors' experimental setup, where Qwen2-VL is used as the policy model and GPT-4o as the external auxiliary model. Furthermore, if this assumption is abandoned, according to the theoretical analysis and experimental conclusions in Luffy [1], directly mixing the trajectories of external auxiliary models with those of the policy model for optimization without additional processing (e.g., calibrating the policy gradient by introducing an off-policy objective function with importance sampling) cannot achieve better results than using only the trajectories of the policy model itself.
- It is risky to simply sort and select the Top-G trajectories from external models and the policy model. The capability of external models is theoretically higher than that of the policy model (which is why external models are introduced), and as a result, it is likely that all trajectories with high rankings in the final selection come from external models. Once the output distribution of external models differs significantly from that of the policy model (which is the case in most scenarios), there will be a risk of training collapse.

[1] Yan J, Li Y, Hu Z, et al. Learning to reason under off-policy guidance[J]. arXiv preprint arXiv:2504.14945, 2025.

**Questions:**

See weaknesses

---

### Official Review · Reviewer_8kCh · 2025-11-02

**Soundness:** 2
**Presentation:** 2
**Contribution:** 1
**Rating:** 2
**Confidence:** 5

**Summary:**

The paper proposes Vision-EKIPL, an RL strategy that introduces responses from external auxiliary models when applying GRPO to the vision-language models (VLMs).
The paper conducts experiments on visual counting, structure perception, and spatial transformation tasks based on Qwen2-VL and show that Vision-EKIPL surpasses Reason-RFT.

**Strengths:**

Vision-EKIPL indeed surpasses Reason-RFT by more than 1 point in the conducted experiments, including three visual reasoning tasks.

**Weaknesses:**

The contribution seems to be small. First, the key modification is to collect a group of actions from external models at each rollout during GRPO. Such a design might be rare in visual reasoning RL, but was sufficiently studied 5 months ago in previous works for large language model (LLM) reasoning [1], and there seems to be no significant differences or challenges when applying a similar strategy to visual reasoning.


Small issues in writing that does not affect the scoring, the citation format hinders the reading of the main text, the paper may consider using \citep instead of \cite.


[1] LUFFY: Learning to Reason Under Off‑Policy Guidance.

**Questions:**

Is there anything special that should be taken care of when introducing the external model-generated actions in GRPO for visual reasoning, which makes it different from the existing similar methods for language reasoning?

---

### Meta-Review · Area_Chair_A4bs · 2026-01-05

**Summary:**

Most of reviewers raise concerns about the lack of comparison between this work and a closely related work LUFFY as they share similar idea. Two reviewers also express concerns about the narrow evaluation setting in this paper, for which limited benchmarks and limited methods are included. One reviewer questions the method's assumption and design. As there is no rebuttal provided, I consider this paper contain obscure contribution and lack sufficient evidence to support their design. Thus, I cannot recommend acceptance for this paper.

**Reviewer Concerns:**

There is no rebuttal for this paper. Thus, all concerns raised by reviewers remain unsolved.

**Reviewer Scores:**

The authors didn't provide a rebuttal and launch any discussions. I consider it unlikely for reviewers to change their scores even if the discussion is not interrupted.

---

### Decision · Program_Chairs · 2026-01-26

Reject